# Pituitary Abscess: A Challenging Preoperative Diagnosis—A Multicenter Study

**DOI:** 10.3390/medicina59030565

**Published:** 2023-03-14

**Authors:** Charles-Henry Mallereau, Julien Todeschi, Mario Ganau, Hélène Cebula, Maria Teresa Bozzi, Antonio Romano, Tuan Le Van, Irene Ollivier, Ismail Zaed, Giorgio Spatola, Beniamino Nannavecchia, Pierre Mahoudeau, Idir Djennaoui, Christian Debry, Francesco Signorelli, Gianfranco K. I. Ligarotti, Raoul Pop, Seyyid Baloglu, Elsa Fasciglione, Bernard Goichot, Caroline Bund, Jeannot Gaudias, Francois Proust, Salvatore Chibbaro

**Affiliations:** 1Department of Neurosurgery, Strasbourg University Hospital, 67000 Strasbourg, France; 2Department of Neurosurgery, Oxford University Hospitals, Oxford OX3 9DU, UK; 3Department of Neurosurgery, Bari University Hospital, 70100 Bari, Italy; 4Department of Neurosurgery, Parma University Hospital, 43051 Parma, Italy; 5Department of Neurosurgery, Dijon University Hospital, 21231 Dijon, France; 6Department of Neurosurgery, Neurocenter of South Switzerland, EOC, 6900 Lugano, Switzerland; 7Neurosurgery Department, Poliambulanza Hospital, 25010 Brescia, Italy; 8Neurosurgery Department, Teramo Hospital, 64100 Teramo, Italy; 9ENT Department, Strasbourg University Hospital, 67000 Strasbourg, France; 10Neuroradiology Department, Strasbourg University Hospital, 67000 Strasbourg, France; 11Internal Medicine and Endocrinology Unit, Strasbourg University Hospital, 67000 Strasbourg, France; 12Nuclear Medicine Unit, Strasbourg University Hospital, 67000 Strasbourg, France; 13Department of Infectious Diseases, Strasbourg University Hospital, 67000 Strasbourg, France

**Keywords:** pituitary abscess, pituitary adenoma, sellar pathology, sellar abscess, nuclear medicine, 18-FDG PET scan

## Abstract

*Background and Objectives:* Pituitary abscess (PA) is a rare occurrence, representing less than 1% of pituitary lesions, and is defined by the presence of an infected purulent collection within the sella turcica. Pas can be classified as either primary, when the underlying pituitary is normal prior to infection, or secondary, when there is associated a pre-existing sellar pathology (i.e., pituitary adenoma, Rathke’s cleft cysts, or craniopharyngioma), with or without a recent history of surgery. Preoperative diagnosis, owing to both non-specific symptoms and imaging features, remains challenging. Treatment options include endonasal trans-sphenoidal pus evacuation, as well as culture and tailored antibiotic therapy. *Methods:* A retrospective multicenter study, conducted on a prospectively built database over a 20-year period, identified a large series of 84 patients harboring primary sellar abscess. The study aimed to identify crucial clinical and imaging features in order to accelerate appropriate management. *Results*: The most common clinical presentation was a symptom triad consisting of various degrees of asthenia (75%), visual impairment (71%), and headache (50%). Diagnosis was achieved in 95% of cases peri- or postoperatively. Functional recovery was good for visual disturbances and headache. Pituitary function recovery remained very poor (23%), whereas the preoperative diagnosis represented a protective factor. *Conclusions*: In light of the high prevalence of pituitary dysfunction following the management of PAs, early diagnosis and treatment might represent a crucial issue. Currently, there are no standard investigations to establish a conclusive preoperative diagnosis; however, new, emerging imaging methods, in particular nuclear imaging modalities, represent a very promising tool, whose potential warrants further investigations.

## 1. Introduction

Pituitary abscess (PA) is a rare but potentially life-threatening condition accounting for 0.2–1% of all sellar lesions [1,2,3]. PAs are generally classified as primary or secondary according to their etiopathogenesis. Primary PAs represent the most common type, accounting for 67% of all cases so far reported, and result from hematogenous bacterial spread to an otherwise normal pituitary gland. On the contrary, secondary PAs occur in only 33% of cases, and represent a secondary infection of pre-existing sellar lesions (i.e., pituitary adenoma, Rathke’s cleft cysts, or craniopharyngioma), with or without a recent history of surgical treatment [1,4]. In those cases of secondary PAs, factors such as the anatomical distortion and subsequently impaired circulation, as well as the altered immunological status, may be considered a pre-existing condition for abscess formation. On the contrary, due to the limited number of cases so far reported and their heterogeneous management, little is known about the risk factors and natural history typical of primary PAs. Furthermore, the international community lacks consensus over well-established criteria for their conclusive preoperative diagnosis. Clinical presentation may be quite variable, ranging from pauci-symptomatic cases to severe compartment syndrome with mass effect on pituitary gland and/or optic apparatus, hence causing different degrees of hypopituitarism, including diabetes insipidus (DI), and visual disturbances [5]. Despite highly sophisticated clinical and radiological investigations, the preoperative diagnosis of PA remains challenging, and recently led to the proposal for a management algorithm for sellar masses suspected of being PAs [6,7]. Early diagnosis and effective treatment constitute the golden path to try and reduce the high prevalence of pituitary dysfunction in patients treated for PAs, which ultimately represents the reason for their overall dismal prognosis [6]. In the present study, the authors report their experience through a multicenter retrospective series of primary PAs surgically treated and followed up for at least 24 months. By complementing our clinical experience with data from the pertinent literature, the aim of this study was to describe the clinical, radiological, and management features of a large population of pituitary abscesses, and to establish a clinical algorithm from diagnosis to management to help/guide clinicians in their routine practice.

## 2. Patients and Methods

We conducted a retrospective review of 84 cases of primary PAs extracted from a prospectively built database of patients undergoing pituitary surgery from 8 neurosurgical centers during a 20-year period (from January 1999 to December 2018). Inclusion criteria were: (1) detection of pus during the surgical procedure, and (2) germs isolated in bacteriological analysis and/or evidence of acute or chronic inflammation on histopathology. Patients’ demographic, clinical, visual, and pituitary function, consisting of full neuro-ophthalmological examination and endocrinological laboratory testing, including standardized basal and dynamic stimulation tests, were assessed at admission, as well as at 3-, 6-, 12-, and 24-months follow-up for each case of PA preoperatively suspected and postoperatively confirmed. All of this information was matched with neuroimaging and nuclear medicine features evaluated by two independent and blinded neuroradiologists/nuclear medicine specialists with focused interest in pituitary imaging. Categorical variables were presented as numbers and proportions. Univariate analysis was performed by analyzing the factors associated with pituitary recovery and were compared using either the chi-squared test or Fisher’s exact test, according to the theoretical numbers. The results are presented as odds ratios (OR) with their 95% confidence intervals. A *p* value < 0.05 was considered statistically significant. Analyses were performed with R software version 4.1.1. R Core Team (2021) (R Foundation for Statistical Computing, Vienna, Austria). R: A language and environment for statistical computing (https://www.R-project.org/, (accessed on 4 January 2023)). This study was conducted according to the Ethical Principles for Medical Research Involving Human subjects stated in the Declaration of Helsinki issued 2004, and its further revisions made in 2008 and 2013. To report our results, we followed the recommendations of the STROCCS (Strengthening the reporting of observational cohort studies in surgery) statement [8]. The study was approved by the IRB of the French National Neurosurgery society (Reference number: IRB00011687).

## 3. Results

### 3.1. Population

A total of 84 patients harboring a primary PA were identified among 11,531 cases of trans-sphenoidal (microscopic/endoscopic) pituitary surgery, indicating that PAs represent 0.7% of all sellar lesions from our database. The diagnosis was established perioperatively in all patients by: identification of ring-enhancement in post-contrast pituitary MRI scans; detection of frank pus at time of surgical evacuation; and laboratory confirmation through bacteriology analysis and/or histopathology assessment (see also Figure 1). Patient demographic and clinical characteristics are summarized in Table 1. Among the 84 patients included and followed up for a minimum of 24 months (range 24 to 54), 32 (38%) were men and 52 (62%) were women, with a mean age of 44 years (ranging from 14 to 71 years). The mean delay between initial onset of clinical signs/symptoms and surgical intervention was 9.2 months (ranging from 2 weeks to 34 months).

### 3.2. Clinical Manifestations

The following main preoperative symptoms were recorded (see also Table 1 and Table 2): (1) variable degrees of generalized asthenia/weakness in 63/84 cases (75%); (2) unspecific, non-localized headache in 42/84 cases (50%); and (3) visual impairment in 60/84 cases (71%), including bitemporal hemianopia, visual acuity dysfunction, or extrinsic ocular motor dysfunction in 32/84 (38%), 26/84 (31%), and 2/84 cases (2%), respectively. Preoperative anterior pituitary insufficiency was demonstrated in 61/84 cases (73%), including: panhypopituitarism in 28/61 cases (46%); isolated corticotropic insufficiency in 8/61 cases (13%); isolated thyrotropic insufficiency in 6/61 cases (10%); combined corticotropic and thyrotropic insufficiency in 10/61 cases (16%); combined corticotropic and gonadotropic insufficiency in 4/61 cases (6%); and isolated gonadotropic insufficiency in 5/61 cases (8%). Patients with gonadotropic hormonal impairment clinically manifested their dysfunction through various degrees of libido reduction and sexual impotence in 7/61 (11%) and 2/61 cases (3%), respectively. DI was found in 21/84 cases (25%).

### 3.3. Infectious Syndrome

Among the 84 patients, only 7/84 (8%) patients presented with high fever and only 3/84 (3.5%) had frank hyper-leukocytosis (values above 15.0 × 10^9^/L). In addition, the rigorous clinical history taken from all patients revealed that 4/84 cases (5%) had a positive history for dental infections, including granulomatosis.

### 3.4. Radiological Features

CT and MRI scans of all patients showed round shaped or polylobate space occupying lesions (SOL) occupying the sella turcica. Those SOL were fully cystic, partially cystic, or predominantly solid, with an average cranio-caudal and horizontal diameter of 15.5 mm (ranging from 10 to 35 mm) and 11.5 mm (ranging from 9 to 22 mm), respectively. All SOL showed a typical ring enhancement after contrast administration.

MRI DWI sequences alone were not showing a specific restriction except in 1/84 cases (1.1%). Only in 8/84 cases (9.5%) presenting with DI and a cystic ring-enhancing lesion, further metabolic imaging with 18-FDG PET was performed and revealed intense uptake strongly favoring a diagnosis of PA (see also Figure 1).

### 3.5. Bacteriological and Histological Analysis

Among the 84 patients included in this study, a bacterial agent was identified in only 21 (25%) of them. The bacterial agents were frequently Gram-positive, such as *Staphylococcus Aureus* in 9/21 cases (42.8%), and various *Streptococcus* species in 5/21 cases (23.8%). Less frequently, Gram-negative germs, such as *Escherichia coli* and *Pseudomonas aeruginosa* were found in only 3/21 cases (14.2%); similarly, low incidence of fungal agents was found in only 2/21 cases (9.5%). Finally, multi-bacterial agents were identified in 2/21 cases (9.5%). Among the 63/84 cases (75%) showing negative bacteriology, histopathology examination allowed the final diagnosis demonstrating either an acute or a chronic inflammatory process without cellular debris.

### 3.6. Management

All patients underwent endonasal trans-sphenoidal surgery using a macroscopic or endoscopic approach; surgical treatment was always followed by immediate broad-spectrum antibiotic therapy, subsequently converted to targeted antibiotic treatment in 21/85 cases (25%) when responsible bacteria/fungi could be identified. All patients received intravenous (I.V.) cephalosporin or a combination of ceftriaxone and vancomycin for 2 weeks, later switched to oral administration for a further 2 to 4 weeks, as per the advice of our infectious disease specialists.

A specific antibiotic/antifungal treatment course was tailored to the needs of each patient. For instance, when *Staphylococcus aureus* was found, the initial I.V. broad-spectrum antibiotic was switched to oral Penicillin for a further 4 weeks. A similar approach was adopted in the case of *Streptococcus* infection, whereas in more complex cases, such as multi-bacterial and/or multi-resistant agents’ infection, a more complex antibiotic treatment was adopted which included rifampicin, metrodinazole, teicoplanin or clindamycin. Finally, the two cases of fungal infection were treated with amphotericin B.

### 3.7. Outcome

All patients showed a good response to the treatment protocol described above. Headache resolved in all 84 patients within 2 weeks (range from 1 to 4 weeks). All 60 patients presenting with various degrees of preoperative visual impairment improved remarkably after surgical evacuation, and only 4/32 patients (12.5%) showed some residual unilateral superior quandrantanopia. Radiological outcome was also very satisfactory: MRI scans at 3, 6 and 18 months showed a complete regression of the PA in 72/84 cases (86%), whereas in 12/84 cases (14%) the PA appeared to be markedly reduced in volume and with a negligible enhancing component. Only in 3/84 (3.5%) patients was a PA recurrence recorded within a mean delay of 14 weeks (range 9 to 21 weeks) and managed by re-do surgery followed by prolonged antibiotic treatment leading to complete clinical and radiological resolution at the latest follow-up. More than half of patients (11/21 cases) suffering preoperative DI showed complete resolution within 12 weeks (range 3 to 16 weeks). Among the 61/84 patients suffering preoperatively of various degrees of anterior pituitary insufficiency, only 14/61 patients (23%) showed resolution of such dysfunction; the other 47/61 patients (77%) required a permanent hormonal replacement. Variables associated with permanent hormonal replacement in univariate analysis showed a significant protective result in favor of preoperative diagnosis (*p* < 0.01), and a significant risk factor with preoperative panhypopituarism (*p* = 0.04) (see also Table 3).

## 4. Discussion

### 4.1. Results and Outcome

This study identified 84 patients harboring a primary PA among 11,531 pituitary surgeries performed over a period of 20 years in eight neurosurgical centers. PAs represented 0.7% of all pituitary gland pathologies. This finding is consistent with previous data from the literature which estimates an incidence of 0.2 to 1.1% of pituitary/sellar lesions [3,6,9,10,11,12,13,14,15,16,17,18,19,20,21,22]. Similar to what is previously described, we also found a gender predilection, with women being more affected than men [6]. Among the 84 patients included in this retrospective analysis, generalized asthenia appeared to be the most common clinical symptom, affecting 75% of patients who also had an underlying corticotropic and thyrotropic pituitary insufficiency. The second most frequent complaints were visual impairment and headache, in 71% and 50% of patients, respectively: these features are in line with data available from the literature and contributed to the commonly recognized clinical triad of PAs [3,11,12]. DI was also present in 25% of patients, and represented a key feature in the differential diagnosis of PAs, as previously suggested by Liu et al. [13]. More recently, Gao et al. reported that DI can be present in up to 49.8% of patients [20]. Of note, all clinical and laboratory markers of infection did not appear to be helpful in achieving a diagnosis: fever and hyper-leukocytosis were detected only in 8% and 3.5% of patients, respectively, compared with previous figures indicating that 18–49% and 40% of PAs tend to have such presentation [3,11,12,13,20]. Clearly, the presence of clear septic state in the context of a sellar mass lowers the threshold for suspecting a primary or secondary PA and for timely treating of it [7]. Similar to what is previously described, a bacterial or fungal agent could be isolated in only 25% of cases from our series, and the lack of pathologic agent identification represents the main reason to support the start of an empiric antibiotic therapy in the immediate postoperative period in all patients with purulent sellar collections [6,7,14]. Although a positive microbiology test in only one out of four patients could appear low, such rate is actually higher than in previous reports [3,11,12,13]. Many factors had been previously considered accountable for the high rate of inconclusive microbiological testing, including empiric preoperative antibiotic therapy, standard pre-operative prophylaxis, or even technical problems with specimen processing [6]. There were indeed several publications in the literature suggesting that a “sterile” pus collection could not be considered as an abscess and that the diagnosis of Rathke cleft cyst should be favored instead [4,17,18]. On the other hand, Gao et al. included in their series all cases presenting intraoperative pus with postoperative either histopathological or bacteriological evidence of abscess, as we did in our large series [20].

A consensus view suggests that the suspicion of a primary PA can be confirmed whenever frank pus is identified in the sella turcica and a clear inflammatory process without evidence of other pituitary pathologies can be seen on histological examination. Since the advent of 16s ribosomal (16s rRNA) polymerase chain reaction (PCR), which allows the detection and identification of bacterial pathogens despite the use of antibiotic therapy before sampling, we do believe that 16s rRNA PCR should be considered systematically when a primary or secondary PA is suspected due to the high rate of negative pus culture. Similar to what has been previously reported, also in our series the most common pathogenic microorganisms belonged to the *Staphylococcus* species and *Streptococcus* species, with a much smaller incidence of multi-bacterial and fungal PAs [3,6,11,12,13,15,19]. As such, broad-spectrum I.V. antibiotics represent the most appropriate treatment course until the results of bacteriologic testing become available. It has also been previously reported that headache and visual disturbances are more likely to improve than hormonal dysfunction; in fact, almost 70% of patients treated for primary PAs are expected to require long-term hormonal replacement [6]. Variables associated with permanent hormonal replacement in univariate analysis showed a significant protective result in favor of preoperative diagnosis (*p* < 0.01) and a significant risk factor with preoperative panhypopituarism (*p =* 0.04). These results suggest the importance of rapid diagnosis and the need to develop multimodal diagnostic tools to achieve it. In our series, the rate of PA recurrence was only 3.5%, which is much lower than the 9% to 12% reported in the literature from studies with similar follow-up lengths [3,9,15]. The relevant literature suggests that a very long FU should be offered to patients with PAs, because recurrences have been observed up to 11 years later [21].

### 4.2. Pathophysiological Mechanism

PA pathophysiology is still unknown for both primary and secondary PAs, even though several hypotheses have been considered [3,6,13]. Atlas et al. reported that either primary or secondary PA could be the result of both hematogenous seeding or direct extension from an adjacent infectious process, as in cases of sphenoid sinusitis, meningitis, or contamination from a CSF fistula [5]. Vates et al., in their series, reported that 16.7% of patients had a history of sepsis [3]. Other risk factors considered included an underlying immune-compromised condition and previous radiotherapy [11,12]. We have identified, in our series, only four patients with a medical history of dental granuloma that could potentially be considered as the primary infectious focus. This said, the clinical outcomes outlined above might be difficult to improve any further if a systematic approach is not taken from the time of initial referral. Of note, in our series the rate of positive microbiology was inversely proportional to the length of clinical history: the shorter the length of signs and symptoms, the higher the chance of identifying a pathogen. This might also indicate that the pituitary dysfunction might result from a longer colliquative rather than a short-lived compressive pathogenesis. All the above suggest that only early diagnosis and treatment could potentially lead to optimal postoperative results.

### 4.3. Perspectives

Despite a growing body of literature in recent years about PAs, their low incidence makes it impractical to create large prospective series, indicating the complexity of creating guidelines to help practitioners in their daily practice. Our results allow us to draw the following conclusions. First, that the presence of DI should raise the suspicion of PA, in the absence of clinical and radiological evidence of other pituitary pathology [7,22,23]. Second, that previous clinical history of chronic infection, recent dental treatments, steroidal treatments, immune-depression, or other co-morbidities exposing to the risk of infections [24], should always put PA among the differential diagnoses considered, even in the absence of fever or hyper-leukocytosis. Third, the absence of restriction on DWI MRI sequences cannot completely rule out the suspicion of PA commonly used for the diagnosis of cerebral abscess: in fact, it seems that the paramagnetic susceptibility effect at the base of the skull could affect signal scattering and cause artifacts due to the vicinity of the cranio-facial pneumatized sinuses [22,24]. The intrinsic limitations of the MRI have pushed us to try other imaging methods and specifically those provided by nuclear medicine. Previous authors have stressed the importance of considering technetium-labelled white blood cells to detect hyper-metabolism as an indicator of intracranial abscesses [25]. Nuclear medicine has also been advocated as an alternative option to conventional MRI in evaluating the skull base, hence our diagnostic strategic choices are not entirely new [26,27]. Shimamura et al. had previously used thallium 201 scintigraphy to diagnose PAs preoperatively [16], however, because thallium 201 scintigraphy is not very specific to the investigation of infectious processes, we have opted for a more robust 18-FDG PET scan, and our results confirm that this investigation provides a very high preoperative diagnostic rate in patients harboring PAs [17,18,19,20,21]. Based on all of those aspects, we have elaborated a clinical algorithm which leverages not only our clinical experience but also previous data from the relevant literature, and has the potential to substantially help clinicians in their routine clinical practice (see also Figure 2). The main difference from the management algorithm previously proposed by Machado et al., is that we have included the use of metabolic imaging to complement the routine neuroradiology work-up and guide surgical management. This said, we certainly agree with their recommendation that any new onset of visual field defects and sepsis should trigger an expedited surgical debulking [7].

### 4.4. Study Limitation and Strength

While the retrospective and non-randomized design of the present study represent its main limitations, this manuscript reports on the largest series of PAs ever described in the English literature. Such point of strength was achieved through the multi-centric set-up of the study, which exploited an accurate and robust, prospectively-built database, which will allow for the generation of new studies in the near future. Additionally, it should be mentioned that the current knowledge gap on this rare pathological entity should be covered by a systematic review of the over 200 cases now described in the literature; a protocol for such study is currently under submission to the PROSPERO database. 

## 5. Conclusions

PA is a rare occurrence. The diagnosis is usually made intra- or perioperatively, through the surgical evacuation of the purulent collection and microbiological/histological confirmation. This study indicates that prompt diagnosis and treatment of PA leads to a favorable outcome, particularly with regard to headaches and visual disturbances. Although there are currently no reliable diagnostic tools for a conclusive preoperative diagnosis, complementing neuroimaging with nuclear medicine investigations could increase the chances of identifying PAs earlier, and offer prompt treatment to patients harboring this challenging pathology.

## Figures and Tables

**Figure 1 medicina-59-00565-f001:**
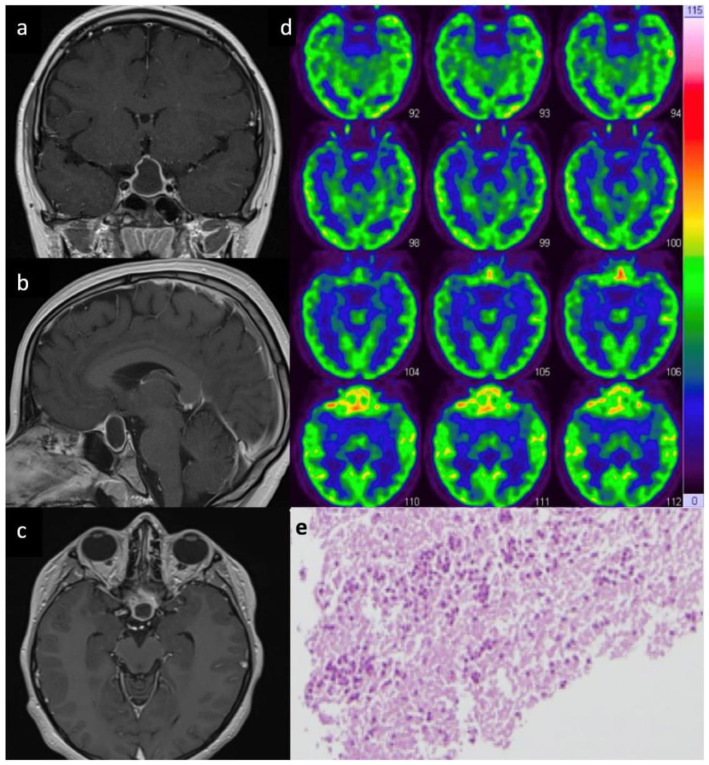
Preoperative T1 weighted enhanced MRI coronal (**a**), sagittal (**b**), and axial (**c**), demonstrating a ring enhancing intra-sellar mass with supra-sellar extension in a 14-year-old boy with a 4 day history of sudden headache, nausea and vomiting associated with thyroid insufficiency. The patient underwent an endoscopic endonasal surgery and the perioperative findings consisted of a primary PA. Bacteriologic investigation revealed a Staphylococcus Aureus infection that was treated by targeted therapy with a favorable outcome. The patient came back 4 months later complaining of progressive headache; an enhanced MRI showed the recurrence of the ring-like intra-suprasellar SOL; on this occasion an 18-FDG PET scan (**d**) showed a hyper-metabolism in the right anterior part of sella, favoring a diagnosis of recurrent PA. Re-do surgery and antibiotic therapy allowed a good outcome without further recurrence at 2-year FU; (**e**) histopathological analysis: the pituitary parenchyma appears fully necrotic and contains an intense pyogenic inflammatory infiltration (hematoxylin eosin × 10).

**Figure 2 medicina-59-00565-f002:**
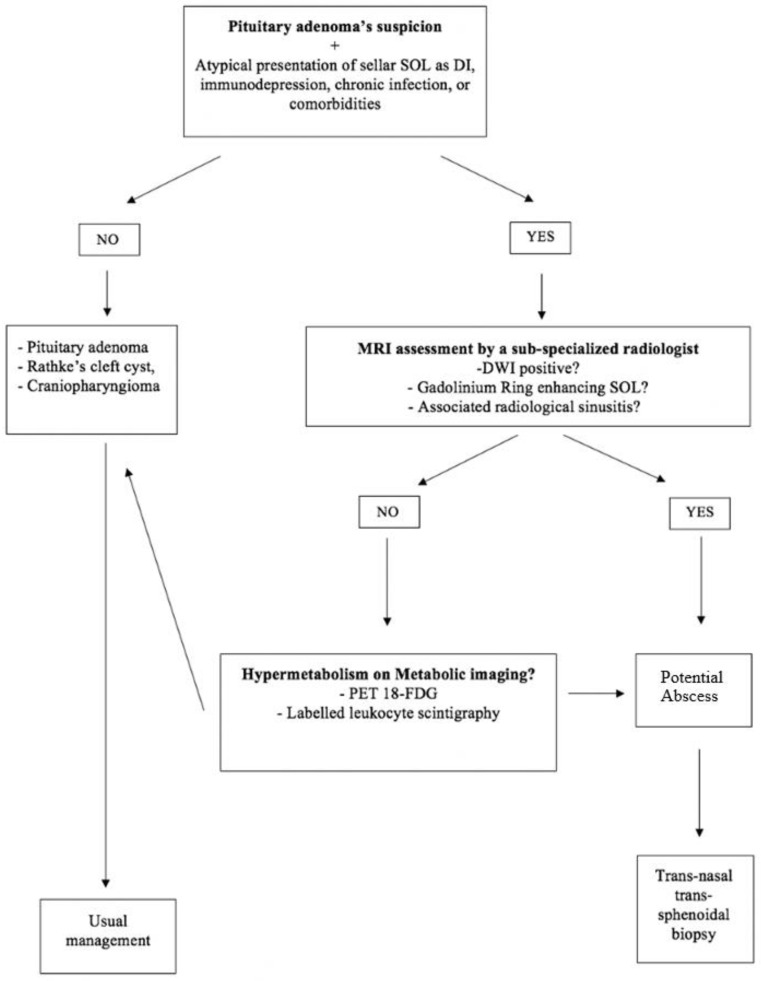
Algorithm for clinical decision-making process in lesions diagnosed as PAs. SOL: space occupying lesion; MRI: Magnetic Resonance Imaging; DI: Diabetes Insipidus; DWI: Diffusion Weighted Images; 18 FDG PET: fluorodesossiglucose positron emission tomography.

**Table 1 medicina-59-00565-t001:** PA patient series: demographic and clinical features.

Characteristics	Patients *n* = 84
Sex ratio (M/F)	0.61
Mean age (years)	44
Mean diagnosis delay (months)	9.2
Main symptoms (%)	
Asthenia	75 (63/84)
Headache	50 (42/84)
Visual disturbance	71 (60/84)
Endocrinological dysfunction	73 (61/84)
Infectious context (%)	
Fever	8 (7/84)
Hyperleucocytosis	3.5 (3/84)
Germs isolated	25 (21/84)
MRI enhanced ^a^ (%)	100 (84/84)
Diagnosis suspected (%)	9.5 (8/84)
Outcome at 24 months (%)	
Persistant headache	0 (0/84)
Persistant minimal visual disturbance	12.5 (4/32)
Persistant hormonal deficit	77 (47/61)
Persistant diabetes insipidus	48 (10/21)
Persistant radiographic lesions	14 (12/84)
Abscess recurrence	3.5 (3/84)

Legends M: male; F: female; a: enhancement after gadolinium injection on T1 weighted images.

**Table 2 medicina-59-00565-t002:** PA patient series: clinical presentation.

Clinical Presentation	Patients (*n* = 84)
General %	
Asthenia	75 (63/84)
Headache	50 (42/84)
Visual disturbance %	71 (60/84)
Bitemporal hemianopsia	38 (32/84)
Visual impairment	31 (26/84)
Oculomoteur paralysis	2 (2/84)
Preop Anterior pituitary insufficiency %	73 (61/84)
Panhypopituarism	46 (28/61)
Isolated CI	13 (8/61)
Isolated TI	10 (6/61)
Combined CI and TI	16 (10/61)
Combined CI and GI	6 (4/61)
Isolated GI	8 (5/61)
Sexual dysfunction	8 (7/84)
Diabetes insipidus	25 (21/84)
Fever	8 (7/84)

Legend: CI: corticotropic insufficiency; TI: thyroid insufficiency; GI: gonadotropic insufficiency.

**Table 3 medicina-59-00565-t003:** Univariate analysis of risk factor determining hormonal deficits.

	Permanent Hormonal Replacement		
Preoperative Characteristics		OR	*p* Value
Women (%)	30/47 (63.8%)	1.32 (0.32, 5.21)	0.65
Anterior pituitary insufficiency (%)			
1 insufficiency	17/47 (36.2%)	3.34 (0.63, 34.27)	0.19
2 insufficiencies	19/47 (40.4%)	2.45 (0.55, 15.51)	0.19
panhypopituarism	25/47 (53.2%)	4.07 (0.92, 25.66)	0.04
Preoperative suspected diagnosis (%)	2/47 (4.3%)	0.06 (0.01, 0.43)	<0.01
Germs isolated (%)	13/47 (27.7%)	0.29 (0.07, 1.18)	0.06

Legend: OR: odds ratio.

## Data Availability

Not applicable.

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
