# Peer review of "Pituitary Abscess: A Challenging Preoperative Diagnosis—A Multicenter Study"

_medicina, 2023, doi:10.3390/medicina59030565_

Round 1

Reviewer 1 Report

This article is a retrospective review of patients who underwent transsphenoidal surgery for pituitary abscess, selected from a prospectively maintained European multicenter database. In total, 84 patients from January 1999 to December 2018 were included. A minimum of 24 months follow-up was an inclusion criterion. As would be expected, many patients had headaches (50%) and varying degrees of pituitary dysfunction (73%) pre-operatively. Diabetes insipidus was also potential indicator of unusual sellar pathology. Infectious signs and symptoms were infrequent. MRI features were varied and not often in keeping with infectious process. The delay between symptom onset and surgery ranged widely, from 2 weeks to 34 months. Ring enhancement seemed to be a radiologic indicator of abscess. Following surgery, headaches improved uniformly, although pituitary function infrequently recovered. The authors conclude: “it is now time to recommend the design of a prospective multi-centric, randomized trial meant to identifying clinical and radiological factors influencing an early pre-operative diagnosis of primary Pas.” I do know what they imply to randomize. If they know it should be specified, otherwise perhaps this sentence should be omitted as it otherwise is meaningless.

The merit of this article is the larger patient population. It is well presented overall. There is no systematic review and others have provided a more in depth discussion. The largest single center experience previously published is by Gao et al (Pituitary 2017) and includes 66 patients. A proper systematic review was performed by Agyei et al (World Neurosurgery 2017), who also outlined in causative organism, clinical presentation, management and outcomes in a more detailed fashion. Nevertheless, the current manuscript complements the existing literature to some degree and adds to the relatively rare published experience over ~>200 cases.

If the data is indeed compiled as described by the authors, an interesting method of displaying data would be to plot recovery of pituitary function with delay between time of symptom onset and surgery.

Author Response

Please find attached a revised version of our manuscript entitled:” Pituitary abscess: a challenging preoperative diagnosis. A multicenter study”, that we are re-submitting for possible publication on Medicina®. We have appreciated all the comments made by the reviewers and undertaken a revision of our manuscript to meet their requests and address their queries. Overall, we feel that their suggestions gave us the opportunity to significantly enhance the quality of our paper and specify some points requiring clarification.

We describe here all the changes made to our manuscript, which are also highlighted in red in the attached Word file.

Reviewer 1

The authors conclude: “it is now time to recommend the design of a prospective multi-centric, randomized trial meant to identifying clinical and radiological factors influencing an early pre-operative diagnosis of primary Pas.” I do know what they imply to randomize. If they know it should be specified, otherwise perhaps this sentence should be omitted as it otherwise is meaningless.

Answer: The authors thank the reviewer for his/her comment and have taken on board the above suggestion by removing the sentence from the revised manuscript.

The merit of this article is the larger patient population. It is well presented overall. There is no systematic review and others have provided a more in depth discussion. The largest single center experience previously published is by Gao et al (Pituitary 2017) and includes 66 patients. A proper systematic review was performed by Agyei et al (World Neurosurgery 2017), who also outlined in causative organism, clinical presentation, management and outcomes in a more detailed fashion. Nevertheless, the current manuscript complements the existing literature to some degree and adds to the relatively rare published experience over ~>200 cases.

Answer: The authors agree with this valuable comment and have added the lack of systematic review of the over 200 cases reported so far in the literature to the comments listed in the section on the limitations of this study.

If the data is indeed compiled as described by the authors, an interesting method of displaying data would be to plot recovery of pituitary function with delay between time of symptom onset and surgery.

Answer: The authors had thought about this option at time of preparing the original submission and definitely agree with the reviewer that our database is amenable for further statistical analysis (including the suggested plotting), however the consensus view among the investigators was to use the present database to generate new studies in the near future.

Reviewer 2 Report

The article “ Pituitary abscess: a challenging preoperative diagnosis. A multicenter study” is a descriptive retrospective study of a series of cases of primary abscess.

The study is of simple design - it is a description of retrospective series of cases. Only descriptive statistics were used.

The strength of the article is a large number of patients (n=84) with primary pituitary abscess, that were included in the analysis. Vast majority of the reports on this type of disease are case reports or literature reviews.

In my opinion the manuscript is worth publishing in Medicina journal and it does not require major revision.

Comments:

I find that most recently published articles on pituitary abscess are commented in discussion of manuscript (doi: 10.25259/SNI_835_2021. eCollection 2021. ; doi: 10.1016/j.wneu.2021.05.137.; doi: 10.1080/02688697.2021.1967877; doi: 10.1007/s11102-020-01115-2. Epub 2021 Jan 12.) I’d encourage the authors confront their data with those recently published more extensively.

Author Response

Please find attached a revised version of our manuscript entitled:” Pituitary abscess: a challenging preoperative diagnosis. A multicenter study”, that we are re-submitting for possible publication on Medicina®. We have appreciated all the comments made by the reviewers and undertaken a revision of our manuscript to meet their requests and address their queries. Overall, we feel that their suggestions gave us the opportunity to significantly enhance the quality of our paper and specify some points requiring clarification.

We describe here all the changes made to our manuscript, which are also highlighted in red in the attached Word file.

Reviewer 2

In my opinion the manuscript is worth publishing in Medicina journal and it does not require major revision. Comments: I find that most recently published articles on pituitary abscess are commented in discussion of manuscript (doi: 10.25259/SNI_835_2021. eCollection 2021. ; doi: 10.1016/j.wneu.2021.05.137.; doi: 10.1080/02688697.2021.1967877; doi: 10.1007/s11102-020-01115-2. Epub 2021 Jan 12.) I’d encourage the authors confront their data with those recently published more extensively.

Answer: The authors thank the reviewer for his/her comment and suggestion of additional studies to complement our discussion, namely: 1) the low diagnostic value of the MRI diffusion sequence (Kawano et al. ), 2) the possibility of abscesses occurring on underlying pituitary lesions such as rathke's cyst (Aranda et al.), and 3) the rarity of bacteria isolation on the pus (Machado et al.). All suggested references have been added to the manuscript and references.

Overall we feel that the quality of this manuscript has much increased after this revision. All typos have now been corrected and we hope that the reviewers will consider the manuscript appropriate for publication in your esteemed journal. While we would be happy to answer further queries should you have any, we would certainly like to thank you for the attention paid so far in helping us revising and improving our manuscript.